# Capturing chemical reactions inside biomolecular condensates with reactive Martini simulations
Christopher Brasnett [1], Armin Kiani [2], Selim Sami[3], Sijbren Otto[2] & Siewert J. Marrink [1] ✉

Biomolecular condensates are phase separated systems that play an important role in the spatio-temporal organisation of cells. Their distinct physico-chemical nature offers a unique environment for chemical reactions to occur. The compartmentalisation of chemical reactions is also believed to be central to the development of early life. To demonstrate how molecular dynamics may be used to capture chemical reactions in condensates, here we perform reactive molecular dynamics simulations using the coarse-grained Martini forcefield. We focus on the formation of rings of benzene-1,3-dithiol inside a synthetic peptide-based condensate, and find that the ring size distribution shifts to larger macrocycles compared to when the reaction takes place in an aqueous environment. Moreover, reaction rates are noticeably increased when the peptides simultaneously undergo phase separation, hinting that condensates may act as chaperones in recruiting molecules to reaction hubs.

Biomolecular condensates have attracted significant attention in recent years due to their ability to dynamically organise biological systems in both time and space, based on liquid-liquid phase separation (LLPS)[1]. This dynamic organisation is thought to be important for the early stages of protocell evolution, as the phase separation of biomolecules into dense regions allowed for the recruitment of many other molecules - such as RNA - without having to pass through a more rigid cell membrane[2–4]. Several studies have now found that condensates are able to act as so-called 'reaction crucibles', whereby molecules are selectively recruited to change the rate or specificity of chemical reactions[5,6]. For example, condensates have been found to change the rate of enzymatic activity and control cellular redox reactions[7–10]. Additionally, condensates themselves may be modified or controlled by such types of chemical reactions, for example through phosphorylation, which can both form and dissolve condensates[11,12].

As the dynamics of the interiors of complex biomolecular systems are challenging to probe experimentally, computational methods such as molecular dynamics (MD) are useful techniques to provide access to molecular resolution[13–16]. However, as necessarily large systems with slow intrinsic dynamics, condensates are computationally expensive to study in atomistic detail. As a result, coarse-grained (CG) models have found significant application in understanding the formation and dynamics of condensates[17–22].

A popular versatile CG force field is Martini, that maps 2–4 heavy atoms onto a single bead, which are parameterised to represent the chemical nature of the underlying components[23,24]. By combining building blocks together, any type of molecule can be represented on a CG scale, enabling a wide range of molecules and phenomena to be studied bridging both biological systems and materials science[25,26]. In the field of biomolecular condensates, Martini-based models have recently been successfully applied to study LLPS of a variety of biomolecules[27–31].

A recent extension of the Martini force field has been the development of reactive Martini[32]. Following previous developments of titratable Martini, reactive Martini makes use of virtual sites and dummy beads to modulate specificity and direction of the binding of two molecules during a simulation[33,34]. Such an approach can be successfully used to model chemical reactions, as demonstrated for a system consisting of benzene-1,3-dithiols, forming higher order oligomers via disulphide bond formation. Reactive Martini can capture chemical reactions occuring in complex, heterogeneous and even out of equilibrium environments, at time and length scales inaccessible with all-atom models. It can therefore predict the influence that changes in physical characteristics (such as concentration, viscosity, or polarity) have on chemical reactions. The development of CG reactive models for benzene-1,3-dithiol in particular is useful; functionalized with peptides these molecules can form self-replicating macrocycles, which are used in attempts to create a synthetic form of life[35–37]. As condensates are now thought also to have played a role in facilitating chemical reactions during the emergence of life from inanimate matter, studying chemical reactions in the presence of condensates is a significant topic of interest.

In this work, we demonstrate how the Martini force field may be applied to such a problem. First, we show how the standard Martini 3 force

[1]Groningen Biomolecular Sciences and Biotechnology Institute, University of Groningen, 9747 AG Groningen, The Netherlands. [2]Centre for Systems Chemistry, Stratingh Institute, University of Groningen, 9747 AG Groningen, The Netherlands. [3]Kenneth S. Pitzer Theory Center and Department of Chemistry, University of California, Berkeley, Berkeley, CA 94720, USA. ✉e-mail: s.j.marrink@rug.nl

field can be used to reproduce the LLPS behaviour of a short model peptide LFssFL, consisting of two LF dipeptides linked with a disulfide spacer. Secondly, by calculation of the potential of mean force (PMF), we demonstrate that benzene-1,3-dithiols preferentially partition into the condensate phase, which we corroborate with an experimental partitioning assay. Finally, as a proof of concept, we simulate the formation of macrocycles from benzene-1,3-dithiol building blocks inside a biomolecular condensate formed by the LFssFL peptides. We show that the presence of biomolecular condensates affects both the rate of reaction, and the size distribution of the macrocycles observed in silico. Our study opens the way to simulate chemical reactions occurring in heterogeneous environments, providing insights into the important role of biomolecular condensates, both in current life forms and at the origins of life.

## Results

### Synthetic peptide condensates modelled with coarse-grained simulations

Recent work has shown that it is possible to form biomolecular condensates from dipeptides linked by synthetic spacers[2,38]. The nature of the phase separation observed was found to be dependent on both the hydrophobicity of the dipeptide 'sticker' involved, and the solvation free energy of the non-interacting spacer linking the two stickers. Inspired by this, we investigated the phase separation of the LFssFL synthetic peptide using the standard (i.e. unreactive) Martini 3 force field, featuring leucine-phenylalanine as a sticker motif and a disulfide spacer (Fig. 1a). According to the experimental data, peptides with this sequence are soluble at neutral pH, but phase separate at high pH (pH > 8).

Figure 1b, c illustrates the final configurations of simulated systems, starting from a random dispersion of the peptides inside the periodic box. In line with the expectations, the peptides have clearly undergone phase separation at high pH, while the system at neutral pH has not. We can use incoherent scattering functions (ISFs) to quantify diffusion coefficients in these systems, which may be used for this purpose on heterogeneous systems by measuring the correlation of a particle from its initial position[27]. As this behaviour should exponentially decay, the number of decay modes

required to fit the data are descriptive of the number of diffusion regimes present. The data were fitted with both single and weighted double exponential decays using lmfit, and the best model for the fit was determined by comparison of the Akaike information criterion[39]. In the homogeneously mixed system at neutral pH only a single decay was needed to fit the data shown in Fig. S1, while two exponential decay functions were required in the system at high pH that has undergone phase separation (Tables S1, S2). The very fast diffusion of the peptides both inside and outside the condensates—and the high degree of exchange between the regions - demonstrates that we can replicate the pH induced LLPS in this system.

We further investigated properties of the high pH condensate by simulating it in a slab geometry. As Fig. 1d shows, the condensate contains around 17% water by weight, implying the dense phase remains well hydrated. However, the measured water content is notably lower than comparable experimental values, reporting water weight contents of >60%[38]. The discrepancy may arise from the way the peptide backbone is represented in the Martini force field; previous work on IDPs has shown that Martini underestimates its hydration strength[40]. This is confirmed by simulations shown in Fig. S2 where simulations were conducted with the peptide-water interactions slightly increased, correspondingly increasing the proportion of water inside the bulk of the condensate. As the aim of the current work is to provide a proof of concept on simulating chemical reactions inside a condensate, we did not attempt to further improve the behaviour of our peptide model.

### Partitioning of reactive components into condensates

Many of the biochemical utilities of condensates have been attributed to their ability to selectively recruit or partition other biomolecules inside them, to increase local concentrations and enhance rates of biochemical reactions. To quantify the ability of condensates to attract benzene-1,3-dithiol, the reactive component considered in this work, we calculated the free energy of transfer to move a single benzene-1,3-dithiol molecule from the dilute region of the system inside a biomolecular condensate formed from the LFssFL peptide described above. The resulting PMF (Fig. 1e) shows that there is indeed a significant energy gain, of about −9 ± 1 kJ/mol, to moving

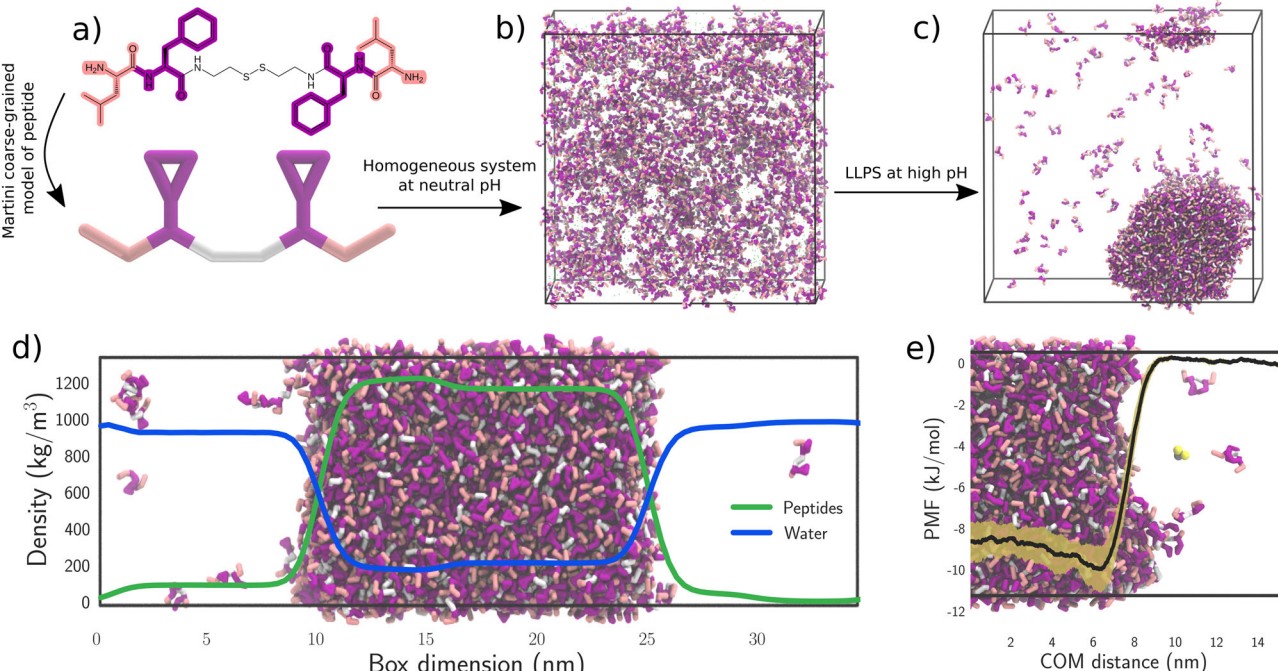

**Fig. 1 | Synthetic peptide condensates. a** chemical structure and CG representation of the LFssFL peptide synthon; **b** no phase separation of the peptide at neutral pH, **c** spontaneous phase separation of the peptide at high pH. **d** density of condensate components at high pH in a slab system. **e** PMF of a benzene-1,3-dithiol molecule from the dense to the dilute phase of the condensate system at high pH.

the molecule within the condensate, suggesting that benzene-1,3-dithiols will spontaneously partition into the condensate. The free energy difference is reduced to around $-7 \pm 1$ kJ/mol in the system with increased water content (Fig. S2), showing that the extent of partitioning will be sensitive to the overall water content of the condensate.

In addition to the PMF calculated for benzene-1,3-dithiol alone, we also calculated the PMF for a single molecule of benzene-1,3-dithiol attached to a short peptide, GLKFK. These molecules (denoted XGLKFK) are known to form extended fibres through chemical reactions forming disulfide bonds, and stabilised by beta-sheet formation between the peptide motifs[37]. The PMF of XGLKFK is shown in Fig. S3, which reveals a similar energetic preference for the condensate ($-10 \pm 2$ kJ/mol). More than the benzene-1,3-dithiol alone, XGLKFK shows a distinct minimum in the interfacial region of the condensates, which suggests that these molecules may accumulate around this region of the condensate before entering the bulk. The minimum is likely as a result of the larger XGLKFK molecule covering a larger interfacial area, and therefore further reducing the surface tension.

To support these simulations, we also measured the partitioning of XGLKFK in LFssFL condensates experimentally (Fig. S11). The partition coefficient of the 1mer between the dense and dilute phase was measured to be $5.6 \pm 0.9$, equating to a free energy of transfer of $-4.3 \pm 0.3$ kJ/mol. In line with the MD results, the experiments show a preference for these molecules to reside in the condensate phase. The quantitative difference in free energies is likely due to (as discussed above) the relative hydrophobicity of the peptide in the current Martini model.

## Macrocycle formation in the presence of condensates

Under experimental conditions, the benzene-1,3-dithiol molecule can be converted into macrocycles as a result of disulfide bond formation[35,41], schematically depicted in Fig. 2a. Previously, using reactive Martini, it was shown that a system initially consisting of only monomers evolves into a system of macrocycles of mostly 3mers and 4mers, in line with experimental results. To see how the presence of a condensate can affect this reaction, we simulated a small LFssFL condensate droplet enriched in the reactants (~94 mM). After 100 ns simulation, we observe that the droplet remains phase separated, containing all of the remaining reactants as well as their products (Fig. 2b, Fig. S4). The progress of the reaction was not observed to affect the formation or structure of the condensate (Fig. S5). The preferential embedding of these compounds inside the condensate is in line with the PMF shown above, indicative of a strong driving force of the reactants to interact with the condensate-forming peptides. Using our experimental assay, we verified that larger macrocycles (3mer and 4mer) indeed also partition into the condensate phase as the monomers do (Fig. S12). We measured partition coefficients of $26.4 \pm 1.3$ and $7.1 \pm 1.2$ for the 3mer and 4mer, respectively, confirming our expectation but indicating that there is a strong and non-trivial cycle-size dependence on the partitioning of these macrocycles. To quantitatively assess the distribution of larger macrocycles within the condensate, we determined the peak area corresponding to various ring sizes in the pre-oxidised precursors, as depicted in Fig. S12, and with partition coefficients summarised in Table S5.

To quantify the reactions that have occurred in our reactive Martini simulations, we obtained the final ring size distribution (Fig. 2c, Table S4, Fig. S6). For comparison, a purely aqueous system with the same overall number of peptides was also simulated. While the most common ring size remains a 3mer, the most noticeable result is the shift towards larger ring sizes when the reaction proceeds inside the condensate. In principle, the observed shift toward larger ring sizes could have a number of reasons. First, the local concentration of the reactants is much higher inside the condensate compared to the pure aqueous case. Second, the rate of the reaction could be enhanced by the presence of the peptides. And third, the larger macrocycles could be thermodynamically favoured inside the condensates. To test the first possibility, we additionally simulated a smaller aqueous system with an increased concentration of the reactants, matching the local concentration inside the simulated condensate. The resulting size distribution indeed

shows a similar shift toward larger macrocycles, suggesting that the concentration effect of the condensate plays a primary role in shifting the equilibrium.

Note that while the increase in ring size is certainly remarkable, we also observe formation of some very large macrocycles of 10–14 monomers. Although there is not yet experimental evidence for such large macrocycles formed in condensates, recent studies have found evidence for large macrocycles formation as a result of directed self-assembly, or high solvent oxidation levels[42,43]. Another aspect that could affect the ring size distribution is the possibility for disulfide bond breaking, which is not captured in our current model. The dissociation energy of disulfide bonds is on the order of 250 kJ/mol[44], and therefore irreversible on the time scale of our simulations, but depending on pH and presence of a reductive environment disulfide bonds can be formed and broken reversibly in practice, e.g. via thiol-disulfide exchange. As a further proof-of-concept of how the reactive potential may be tuned to optimally reflect experimental conditions, we performed additional simulations to demonstrate that with a smaller potential well, bond breakage in this manner can be reproduced (Figs S7, S8). In this case too, very large macrocycles are formed, but are also seen to break apart. Similar to the irreversible case, 2mers and 3mers still remain the dominant ring sizes observed in these systems, appearing to approach an equilibrium population in the last 20 ns of simulation time.

## Macrocycle formation under out-of-equilibrium conditions

In the simulation setups described above we initially placed the reactants inside the condensate, accounting for their strong preferred partitioning into these condensates as suggested by the PMF and the experimental partitioning assay. However, under realistic conditions the macrocyclization might start before the reactants are fully integrated inside the condensates, i.e. occurs under out-of-equilibrium conditions during self-assembly of the condensate. To investigate the co-formation of condensates and macrocycle rings, we simulated six systems of varying size and both peptide and reactive component concentration. Illustrations of initial and final configurations are shown in Fig. S4, and the ring size distributions across three replica trajectories are shown in Fig. 2c. As shown by the counts in Fig. 2c, in each of the conditions simulated the modal ring size is the 3mer, as observed in the pure aqueous system. The most variation is then seen in the maximum size of macrocycles formed by the reactive molecules. Randomly introducing phase separating peptides into the reactive system, the maximum ring size after 100 ns simulation is observed to be 7mer, rather than 5mer or 6mer observed in the reference simulations. The concentration of peptides added to the system only had a slight effect. With 500 peptides—a weight ratio of 2% peptide—neither the distribution nor the maximum ring size changed substantially, while increasing the concentration to either 1500 (6% weight) or 2500 (12% weight) peptides also appeared to stabilise the number of monomers forming a ring. In these co-formation simulations, while the phase separation of the peptides was not completed after 100 ns, the distribution of ring sizes was not subsequently observed to change substantially over the course of 500 ns (Fig. S9). Further, in all simultaneous phase separation simulations, the reacted rings were seen to spontaneously partition inside the condensate droplets that had already formed by 100 ns and eventually merged to a single condensate droplet as illustrated for the system containing 1500 peptides in Fig. 2e. This effect of condensate weight proportion was also observed in a second small peptide-based condensate system, using the WGR-1 peptide[45], results for which are shown in Fig. S10.

While we can use reactive Martini to probe the size and stability of the macrocycles formed by benzene-1,3-dithiol, we can also investigate the rate of the reactions. In Fig. 2d, upper panel, we show how the reactive monomers are 'consumed' as the simulations progress, for the four systems where the condensate is formed simultaneously. The lower panel of Fig. 2d then shows the underlying rate of this consumption, indication how fast monomers are paired up into at least 2mers. Notably, there appears to be a difference in the rates of consumption comparing the systems in water alone or at a low concentration of peptides, versus systems with higher concentrations of peptides. This suggests that the rate of reaction in these

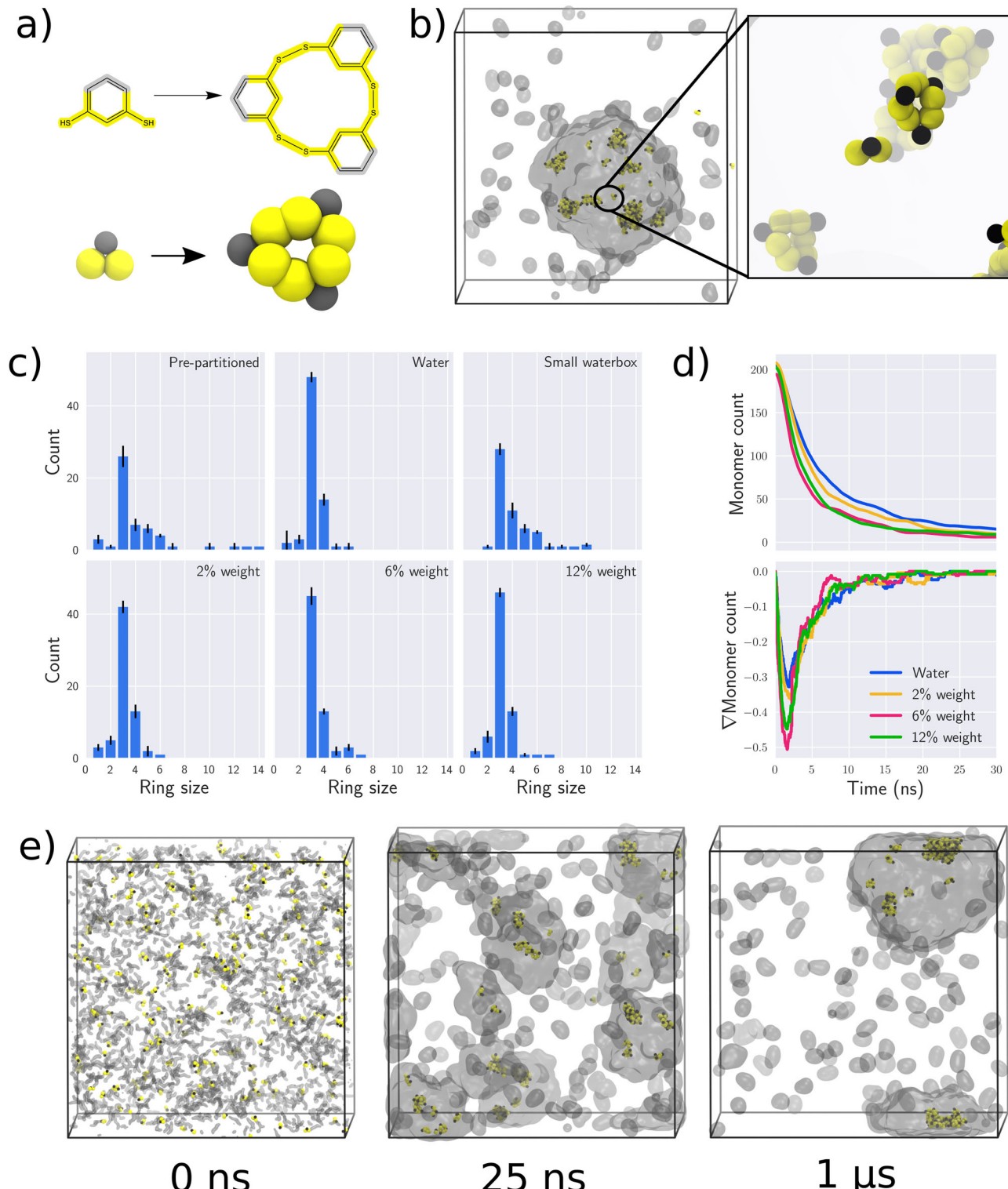

**Fig. 2 | Macrocycle formation in the presence of condensates. a** Oxidation of benzene-1,3-dithiols leads to the formation of rings, illustrated by the chemical reaction and its representation with reactive Martini. **b** Reacted components inside a condensate (shown as a transparent grey surface) from a pre-partitioned initial configuration. The highlight shows two 3mers and an unreacted monomer, with a small aggregate of several rings in the background. **c** Final distributions of ring sizes across the six different conditions investigated. Error bars denote the statistical error from three independent simulations. **d** Monomer consumption (top panel) and the underlying rate (bottom panel) for co-assembling systems of condensates and macrocycles. **e** simulation snapshots of a co-assembling system of 1500 peptides and 216 reactive molecules. Starting from a dispersed configuration, reactions are complete after around 25 ns with reacted components inside small condensates. By 1 μs, individual droplets of condensates have coalesced into a single large one.

systems is coupled to the rate at which peptides also undergo phase separation. As we observe a very strong partitioning of individual benzene-1,3-dithiol into condensates, it would appear that the condensate-forming peptides may act as chaperones for monomers, aiding their partitioning into dense condensates. As the effective concentration of monomers is then increased inside condensate droplets, the rates of reaction are correspondingly increased.

## Discussion and conclusion

In this work, we have demonstrated how reactive Martini can be used to explore disulfide bond formation between benzene-1,3-dithiol monomers leading to the formation of macrocycles inside biomolecular condensates. We firstly demonstrated that the Martini force field can be used to reproduce the expected pH-dependent LLPS of a synthetic peptide. Secondly, we showed that the reactants preferentially partition inside condensates, with partitioning coefficients comparing well between simulation and experiments. Finally, we observed that the condensates formed by the synthetic peptides can change both the speed and the outcome of the reactions between benzene-1,3-dithiol molecules by acting as an attracting environment for the reactants. This effect is dependent on the volume fraction of condensate-forming peptides in the system, which drives the formation of larger macrocycles at higher fractions. Similar volume fraction rate enhancements have been found recently using kinetic network models[46]. In addition, we showed that the reaction products can be sensitive to the detailed environment of the condensate, such as water content and that reversible reactions can be studied by changing the well depth of the reactive potential.

Based on the current proof-of-concept study, we anticipate that Reactive Martini in general can be applied to probe the effect of a large diversity of complex environments on reaction kinetics and products, even in out-of-equilibrium conditions such as during self-assembly. Although we demonstrated this for a single type of reaction only and focused on a particular condensate-forming system, the general nature of the Martini forcefield makes extension to other systems straightforward in principle. To further back up this statement, we considered a second condensate system, using the WGR-1 peptide[45], as shown in Fig. S10. The results obtained reveal a similar effect on the reaction characteristics of benzene-1,3-dithiol, with a shift to formation of larger ring sizes due to the increased concentration of the reactants inside the condensate.

As noted in the original reactive Martini paper, one significant challenge for the further development and utilisation of the model is the use of the group scheme for tabulated potentials. With the current version of Gromacs, the time scales that can be reached with reactive Martini are necessarily limited. However, any future implementation of the Verlet scheme in Gromacs would certainly allow studies such as the present work to be carried out over much longer timescales (100 s of microseconds), and give us more insight into how chemical reactions are affected by different types of biomolecular condensates. Another challenge in this work was to capture the phase separation of biomolecules in tandem with the completion of chemical reactions, as the former takes far longer to complete than the latter. To further facilitate such studies, the reaction barrier in reactive Martini simulations could be optimised to tune the rate of reactions and make reaction timescales more compatible with the phase separation process.

The significant macrocycle sizes observed in this work also suggest that the reactive potential might still require further optimisation to better reflect the nature of the formation of benzenedithiol-based macrocycles. Further extensions to the reactive Martini framework could allow for the peptide side chains of functionalized benzene-1,3-dithiol building blocks to form β sheets on contact, allowing for the stable formation of stacks of rings in silico. Finally, with further development of titratable Martini, the pH dependent dynamics of reactions could also be studied. Indeed, the LFssFL peptide condensate primarily used in this work undergoes phase separation triggered by an increase in pH and the deprotonation of the two N termini. Future studies combining these models with those of biomolecular condensates will further allow us to understand how reaction rates can be enhanced in both primitive biomolecular systems reflecting early-life conditions or in membraneless organelles found in modern day cells.

## Methods

### Simulations

Input parameters for the LFssFL and WGR peptides were generated using Polyply, with standard mappings for Martini amino acids In the LFssFL synthetic peptide, the disulfide spacer was mapped as two SP1 beads[47]. At neutral pH, the two N termini of the LFssFL synthetic peptide were both modelled with Q5 beads both of +1 charge to reflect the expected protonation state. At high pH, the deprotonated termini were both modelled as uncharged P6 beads. Note that these homogeneous protonation states were used to simplistically model the change in pH in order to reproduce the overall phase separation of the system. To increase the interaction between peptides and water for selected simulations, a virtual site was added to the backbone beads of the peptides with an extra interaction of $\varepsilon = 0.465$ nm and $\sigma = 0.1$ kJ/mol to the water beads of the system, as detailed elsewhere[48].

The simulation input topology for unreactive benzene-1,3-dithiol was taken from the original reactive Martini paper, using the model without its reactive components. The model of XGLKFK was constructed with bonded parameters based on the model of ref. 37. Parameters for reactive benzene-1,3-dithiol were also directly used from the reactive Martini paper, which parametrised the reaction bond, angles and dihedrals from atomistic simulations, which in turn was parametrised by quantum mechanical calculations using Q-Force[32,49]. For simulations using a smaller potential well, the notebook provided with reactive Martini was used to write new tabulated potential files with a reactive potential reduced from 60 kJ/mol to 20 kJ/mol. Systems were prepared using standard tools in Gromacs, apart from the pre-partitioned system, where the gen_coords program of Polyply was used to place molecules within a spherical geometry. System details are listed in Table S3.

Simulations were performed with Gromacs 2018.8, required for the use of tabulated potentials in reactive Martini simulations[50]. For simulations of peptide systems alone using non-reactive Martini, the standard set of Martini simulation parameters after de Jong et al. were used, with non-bonded Lennard-Jones and Coulombic interactions cutoff at 1.1 nm[51]. The temperature was maintained at 300 K using the velocity rescaling thermostat with a coupling parameter of 1 ps[52]. Pressure was maintained at 1 bar using the Berendsen (coupling time 4 ps) and Parrinello-Rahman (coupling time 12 ps) barostats for equilibration and production simulations respectively, with a compressibility of $3 \times 10^{-4}\,\mathrm{bar}^{-1}$ [53,54]. The integration timestep was 10 fs and 20 fs for equilibration and production simulations respectively. PMFs were computed using the accelerated weight histogram (AWH) method along a reaction coordinate[55]. The reaction coordinate was defined as the distance between the centre of mass of a pre-equilibrated condensate slab, and the benzenedithiol molecule. PMFs were averaged over 3 independent simulations of 1 μs each.

In reactive simulations, the parameters developed especially for reactive Martini were used[32]. For the non-reactive components of the system, the run parameters are identical to the 'standard' Martini parameters with nonbonded interactions cutoff at 1.1 nm. To enable the use of tabulated potentials, the group cutoff scheme was used with a neighbour list size of 1.2 nm, which was updated every 10 steps. An integration timestep of 10 fs was used for both equilibration and production simulations. Additionally, a LINCS order of 8 with 2 iterations was used to solve constraints. Production simulations were performed in triplicate.

### Experiments

All reagents, solvents and buffers were purchased from commercial sources and used without further modifications. Building block XGLKFK was synthesised by Cambridge Peptide Ltd. (Birmingham, UK) and the Spruijt lab (Radboud University, Nijmegen, Netherlands) provided us with the condensate material (LFssFL).

To determine the partitioning of different species in the condensates, samples of LFssFL condensates with final concentration of 3 mM were prepared at pH 8.2 (borate buffer, 200 mM) in conical glass vials. Then 50 μL of 1mer and 175 μL of 3mer and 4mer were spiked into the solution to have a final volume of 500 μL and final concentration of 0.2 mM for all species. These samples were incubated for 30 min at 4 °C to decrease the side reactions. Then 60 μL of each sample (original solution) was removed and injected to UPLC to measure the total concentration of each component. The glass vials were centrifuged at 5000 rcf for 30 min at 4 °C. After centrifugation, 60 μL of the supernatants were taken immediately from the top of solution and analysed by UPLC to measure the concentration of each component in the supernatant phase. The volume of the remaining supernatant was then measured precisely by standard micropipette (error = 1 μL) and the volume of condensate phase was calculated by difference of total volume (440 μL) and supernatant volume.

Replicator building blocks (1mer) can form hybrid products with LFssFL. To prevent the formation of hybrid products, we preoxidized 1mer to generate precursors (3mer and 4mer) while retaining a minimal amount of 1mer to facilitate thiol-disulfide exchange.

Three independent repeats of each component and the standard solutions were prepared. For standard solutions, different concentration of each component (0.05 mM, 0.1 mM, 0.15 mM and 0.2 mM) were prepared in buffer and a calibration curve was constructed. In order to calculate the concentration of each component in the supernatant and condensate phase, the average peak area of each component was compared to the average peak area of the standard solutions and, based on the calibration curve, the concentration of each component was measured in the solution before centrifugation and supernatant phase. The concentration in the condensate phase was calculated from the difference of concentration as following:

$C_{condensate} = (C_{total} - nC_{supernatant})*(V_{total}/V_{cocarvate})$

$C_{condensate}$ : concentration of component in the condensate phase

$C_{total}$ : concentration of component before centrifugation

$C_{supernatant}$ : concentration of component in the supernatant

$V_{total}$ : total volume

$V_{cocarvate}$ : volume of condensate phase

n: fraction of supernatant ($V_{supernatant}/V_{total}$)

Finally the partitioning of each component was calculated as the ratio of the concentration of that component in the condensate phase and in the supernatant.

## Data availability

Data are available from the authors on request. Essential simulation and analysis files are available at the Zenodo repository (https://doi.org/10.5281/zenodo.12188970).

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

## Acknowledgements
We thank the Center for Information Technology of the University of Groningen for their support and for providing access to the Hábrók high performance computing cluster. We also thank Jan Stevens for useful discussions throughout this work, and Maria Tsanai for initial work on the LFssFL condensate systems. We are grateful to the lab of Evan Spruijt for providing LFssFL synthetic peptide for our experiments. The work presented in this article is supported by the Novo Nordisk Foundation grant NNF20OC0063808, 'BOUNDLESS'.

## Author contributions
S.J.M, C.B., A.K., S.O. designed the research. C.B. performed the simulations and analysed the data with advice from S.J.M. and S.S. A.K. performed the experiments and analysed the results. C.B. and S.J.M. wrote the manuscript with input from all authors.

## Competing interests
The authors declare no competing interests.
