## [Peer Review File · Communications Chemistry]

Reviewers' comments:

Reviewer #1 (Remarks to the Author):

Brasnett et. al describe the use of the reactive Martini framework (developed by the same group in an article earlier this year, Sami & Marrink, J Chem Theory Comp 2023) to analyse a chemical reaction taking place in a phase-separated mixture. The reaction is the same as in the previous study: the formation of macrocycles via disulfide bond formation between benzenedithiol rings. The coacervate host is formed by the simple coacervation of a minimal peptide, LFssFL, introduced by Abbas et al in Nat Chem 2021. The authors first show that the Martini 3.0 force field can be used to describe this minimal phase-separating system. Then they combine the two elements to investigate the course of the macrocycle-forming reaction in the presence of the LFssFL condensates. Their main conclusion is that the condensates increase the reaction rate due to an increase in local reactant concentration. In addition, they found a subtle tendency to form larger macrocycles in the presence of condensates.

Their question is of great interest to the condensates/coacervates community and therefore would fit well in the themed collection. However, in its current state the manuscript's discussion comes off as superficial and incomplete, and I would like to recommend some additional experiments and deepening of the discussion before supporting its publication. The main conclusion about the effect of concentration is largely expected, and it would add more value if it was demonstrated for multiple pairs of coacervates and reactions. For example, this would allow the authors to establish a clear correlation between partitioning coefficients and reaction rate enhancement. The second conclusion, the difference in product distribution, is based on a subtle variation, and lacks other examples to support it.

I believe that the impact of this manuscript depends on the possibility to generalize their MD method to other systems, and show that it can capture differences between reactions. I am particularly worried that the conclusions are built over a single reaction-coacervate pair without a comparison to experimentally measured reaction rates, and may not be general, or at least they don't demonstrate the capabilities of the MD method. The authors' own conclusions point out several further developments needed, and I wish some were already addressed in this manuscript.

Point by point comments:

Introduction

- Could you elaborate on the sentence "As the interiors of complex biomolecular systems are challenging to probe experimentally"? Why or when are microscopy, HPLC/NMR, not suitable?

- I am missing some of the reasoning behind the model system. Please explain: a) the choice for the LFssFL peptide, given that in the reference used there were other coacervate-forming options such as FFssFF and LLssFF? b) the need/advantage of using the coarse-grained Martini force field in the case of such small molecules (in comparison for example to atomic force fields such as FF99IDPs, recently used for hexapeptides by Nicy et al in QRB Discovery 2023).

- Could you explain (to an experimentalist) what are the features captured by the Reactive Martini framework and how? What effects is it expected to capture? Can it predict the influence of parameters other than the local concentration of reactants, which is already known? For example, the influence of diffusion coefficients, dielectric constant/polarity of the medium etc?

- This sentence is difficult to read with the two consecutive gerunds:

"As condensates are now thought also to have played a role in the emergence of life from inanimate matter, providing reactive compartments, studying chemical reactions in the presence of condensates is a significant topic of interest."

- Could you make it clear that coacervates and condensates are used interchangeably? It would be interesting to include a discussion on the differences in simulating complex coacervates and simple coacervates like the ones in this paper.

Results and Discussion

- Some sentences are written in a very casual style, assuming the reader is very familiar with the topic and the project. For example

"Recent work by the Spruijt group has shown"

"Parameters for unreactive benzene-1,3-dithiol were taken from the original reactive Martini paper"

- As you say, "the number of decay modes required to fit the data are descriptive of the number of diffusion regimes present". Please provide a measure of the quality of the fitting for each case (R2 if suitable).

- The discrepancy between experimentally and computationally obtained water content is quite large (60% Vs 17%) and I do not understand how it does not compromise the analysis of the effect of the coacervate environment. I would assume the dielectric constant of the medium to be relevant for the disulfide formation, and then be very different if you consider 60 or 17% of water.

To address this, I suggest: including experimental data for this reaction to show the quality of the simulation's prediction. If that is not possible, include a comparison of simulation and experiment for a second system for which it is possible - the reaction is slower, or that can be monitored with fluorescence microscopy. Since you focus on the effect over the reaction rate, a comparison to experimental data is really missing. Alternatively, please include simulations with other phase-separating peptides and perform reactive Martini with the one with the best water content simulation data.

- In the Partitioning section, I could not find the experimentally measured Kp for the 1,3-benzenedithiol. Please include all in a table - measured and calculated Kp and free energy of all compounds studied (including macrocycles).

- I did not understand the reasoning behind choosing the XGLKFK as a second guest molecule for the coacervates - a more similar variant of the benzenedithiol would be more helpful to understand what factors influence partitioning. And more reaction models would be necessary to establish a clear correlation between partitioning coefficients and reaction rate.

- the discussion about Figure 2 is not clear (starting on "In the previous setup..."), the naming of the six experiments could also be clearer. Since this is one of the two main conclusions of the manuscript, it needs to be indicated whether the difference between the product distributions is statistically significant. The results with water, 500, 1500 and 2500 peptides seem equivalent and I can't tell what variation is significant in MD simulations.

- does the benzenedithiol reaction affect the phase separation of the peptide? Or does phase separation take place the same way in your co-formation simulations (500, 1500 and 2500 peptides) as in your non-reactive Martini ones?

- the interfacial partitioning is an interesting finding, please discuss whether it affects the reaction. Could it explain the difference observed for pre-partitioned dithiol and co-formation simulations - I noticed the size of the condensate varies on each case, and so does the interfacial area.

- Could you clarify if the monomer and products count refer to the whole box, or only to the coacervate phase? Please provide the values of the count and the reaction rate on a table, separating total, dilute phase and coacervate phase - this is important to give a full picture of the reaction progress.

- When you say "In principle, the observed shift toward larger ring sizes could have a number of reasons", you list three possibilities - concentration, peptide catalysis and thermodynamic stability of the products. You attribute the effect entirely to concentration, but it's difficult to agree with that unless you include other coacervate-reaction pairs, or test for the other factors too - is that something you can see with reactive Martini?

- Since the coacervate-forming peptide is a disulfide itself, do you observe hybrid products from

the thiol-disulfide exchange between LFssFL and benzenedithiol experimentally? Or do you take it into account computationally? This might affect effective concentrations, or the assumption that the peptide is non reactive.

- I think a big question in the field is not only what happens to reactions inside coacervates, but what happens to kinetics when the reactants are involved in phase separation. Can you include a simulation of the coacervation and the reaction for LFssFL alone, starting from LF-SH? This is will be a natural expectation of the active coacervates community. Otherwise, please discuss the challenges involved in making that step.

Conclusions

- In general, the conclusions are too focused on further developments and should elaborate more on the findings of the paper itself.

- In the conclusion "We observed that the condensate can change both the speed and the outcome of the reactions by acting as an attracting environment for both the reactants and the products", can you explain why the partitioning of the products plays a role in the reaction rate? I did not find that claim in the discussion.

Methods

- If you are measuring the volume of the dilute phase precisely, why do you consider it in the equation that the total volume and the dilute phase volume are the same? Or could you include a table with the volume measurements?

Reviewer #2 (Remarks to the Author):

Brasnett et al. investigate the impact of biomolecular condensate formation on the rate of chemical reactions using the reactive Martini model. Specifically, they explore how condensates composed of simple peptides, previously studied through experiments, can accelerate macrocycle formation. The manuscript is well-written, clear, and concise. The experimental data (i.e., spectral data) do support the findings in the paper.

Key Finding:

1. The concentration of reactive molecules within the condensates exerts the strongest influence on the size of formed macrocycles. These results carry significant implications for understanding the role of condensates in reaction kinetics.

Minor Comments:

1. Additional details or annotations in Figure 1 would enhance its accessibility to a broad audience.
2. Reporting the mass density or concentration of components in Table S2 would be beneficial.
3. Including mean and standard deviation values in Figure 2c would improve the clarity of trends.
4. In the conclusion section, a comment comparing the reported rates to experimental findings would provide better contextualization of the key findings.

Reviewer #3 (Remarks to the Author):

In this work, titled "Capturing chemical reactions inside biomolecular condensates with reactive Martini simulations" the authors present a proof-of-concept study into the application of the Martini 3 model to the formation of macrocycles of benzene-1,3-dithiol inside a condensate made of synthetic peptides. I really enjoyed reading this manuscript. I think the work represents a valuable contribution to the field. It is very exciting to see the first steps in developing and adapting the powerful Martini model towards probing chemical specificity and chemical reactions inside condensates. I would be happy to recommend publication of this work, but have a few comments and questions for the authors:

1. The authors explore, qualitatively, the impact of pH (low versus high pH) on the phase

behaviour of LFssFL solutions with Martini 3. To mimic the effects of pH, they change the charge of the peptides to account for the change in the protonation state at low and high pH; then they run DC simulations to calculate the density profiles across the long side of the box in the slab geometry. These profiles show phase separation of the solution into a LFssFL-rich and water-rich coexisting phases at high pH, and a homogeneous solution at low pH, in perfect agreement with the experiments of the Spruijt group. However, the approximation used to mimic the impact of pH is too crude, albeit likely necessary since accounting for the impact of pH on simulations of biomolecular phase separation is extremely challenging—e.g. pH changes are expected to yield a multiplicity of interconverting molecular states with varying protonation states. I think the paper would benefit from an explanation of why the approximation used by the authors is reasonable. What are the difficulties with describing the impact of pH on the phase behaviour of biomolecules? What are the pKas of the dipeptides? And Why is assuming homogeneous protonation states adequate in this case?

2. The authors discuss that the Martini condensates contain ~17% of water, while experiments have reported a much higher water weight of >60%. Then, they explain that this is due to the Martini underestimating the strength of interactions between amino acids and water. After this, they introduce their results of umbrella sampling simulations to quantify the PMF of moving a benzene-1,3-dithiol molecule (and the molecule attached to peptides) from the LFssFL-rich phase to the water-rich phase. The PMFs suggest spontaneous partitioning of benzene-1,3-dithiol into the LFssFL-rich phase with a small energy difference of around -9 kJ/mol. They also observe a minimum in energy near the condensate interphase, suggesting preferential accumulation of the small molecule near the interphase. However, I would expect such an energy difference to be heavily reduced and the shape of the PMF curve to change if the interactions of LFssFL with water in the model were to be increased to form condensates with a higher water content, as suggested by experiments. Can the authors comment on this?

3. The next part of the paper is very exciting as it demonstrates the application of Martini to probe chemical reactions inside condensates, particularly the formation of macrocycles from monomers of benzene-1,3-dithiol. However, a big drawback that I see is disclosed by the authors: their algorithm only allows for the formation, but not breakage of the macrocycles. This implies that the forces in their system are non-conservative, and the system is out of equilibrium. Thus, the preferential cycle sizes they observe (Fig 2c) and rates of reactions are simply determined by how long the simulation has been run, or can the authors describe steady states?

Minor:

1. The term "coacervate" is used incorrectly. Coacervate refers to a condensate formed by at least two components via associate heterotypic interactions (e.g polyR and RNA).

We thank all the reviewers for their thoughtful comment on our manuscript. We provide our responses in red and where applicable, changes to the text in blue

Reviewers' comments:

Reviewer #1 (Remarks to the Author):

Brasnett et. al describe the use of the reactive Martini framework (developed by the same group in an article earlier this year, Sami & Marrink, J Chem Theory Comp 2023) to analyse a chemical reaction taking place in a phase-separated mixture. The reaction is the same as in the previous study: the formation of macrocycles via disulfide bond formation between benzenedithiol rings. The coacervate host is formed by the simple coacervation of a minimal peptide, LFssFL, introduced by Abbas et al in Nat Chem 2021. The authors first show that the Martini 3.0 force field can be used to describe this minimal phase-separating system. Then they combine the two elements to investigate the course of the macrocycle-forming reaction in the presence of the LFssFL condensates. Their main conclusion is that the condensates increase the reaction rate due to an increase in local reactant concentration. In addition, they found a subtle tendency to form larger macrocycles in the presence of condensates.

Their question is of great interest to the condensates/coacervates community and therefore would fit well in the themed collection. However, in its current state the manuscript's discussion comes off as superficial and incomplete, and I would like to recommend some additional experiments and deepening of the discussion before supporting its publication. The main conclusion about the effect of concentration is largely expected, and it would add more value if it was demonstrated for multiple pairs of coacervates and reactions. For example, this would allow the authors to establish a clear correlation between partitioning coefficients and reaction rate enhancement. The second conclusion, the difference in product distribution, is based on a subtle variation, and lacks other examples to support it.

I believe that the impact of this manuscript depends on the possibility to generalize their MD method to other systems, and show that it can capture differences between reactions. I am particularly worried that the conclusions are built over a single reaction-coacervate pair without a comparison to experimentally measured reaction rates, and may not be general, or at least they don't demonstrate the capabilities of the MD method. The authors' own conclusions point out several further developments needed, and I wish some were already addressed in this manuscript.

We thank the reviewer for their substantial and thoughtful comments on the manuscript. We have responded to their comments point by point below. We would like to stress, however, that the current manuscript is not meant to be a study on the effect of condensates on reactions in general, but in the first place a proof of concept that this process can be captured with MD simulations. Nevertheless, we agree that it would strengthen the paper to have more examples, and therefore performed additional simulations (i) using a different condensate system, (ii)

exploring the effect of water content of the condensate, and (iii) changing the reaction scheme to allow for reversible disulfide bond breaking.

Point by point comments:

Introduction

- Could you elaborate on the sentence "As the interiors of complex biomolecular systems are challenging to probe experimentally"? Why or when are microscopy, HPLC/NMR, not suitable?

Here, we mean that understanding the chemical detail and interactions within the bulk regions of biomolecular condensates is challenging, and probing the complex dynamics of the bulk regions in detail is only really possible using computational approaches such as molecular dynamics. Our overarching aim in the present work is to demonstrate that reactive Martini can be used as a method to investigate these phenomena. To try to clarify our point here, we have changed the sentence to read:

As the dynamics of the interiors of complex biomolecular systems...

- I am missing some of the reasoning behind the model system. Please explain: a) the choice for the LFssFL peptide, given that in the reference used there were other coacervate-forming options such as FFssFF and LLssFF? b) the need/advantage of using the coarse-grained Martini force field in the case of such small molecules (in comparison for example to atomic force fields such as FF99IDPs, recently used for hexapeptides by Nicy et al in QRB Discovery 2023).

The model system was selected because a) this is the same peptide that is principally used for the complementary experimental studies that we have included, and b) the focus of the present work is particularly how the reactive Martini force field can be used with other models of the native force field. Naturally, coarse-grained methods can also extend both the sizes of systems simulated, and the time scales accessible. Simulating the spontaneous formation of large condensate systems using atomistic force fields as we do here would otherwise be prohibitively computationally expensive.

- Could you explain (to an experimentalist) what are the features captured by the Reactive Martini framework and how? What effects is it expected to capture? Can it predict the influence of parameters other than the local concentration of reactants, which is already known? For example, the influence of diffusion coefficients, dielectric constant/polarity of the medium etc?

We have added the following text to the introduction to help clarify these questions:

Reactive Martini can capture chemical reactions occurring in complex, heterogeneous, and even out of equilibrium environments, and is advantageous for scaling with system time and timescales inaccessible with all-atom models. It can therefore predict the influence that changes

in physical characteristics (such as concentration, viscosity, or polarity) have on chemical reactions.

- This sentence is difficult to read with the two consecutive gerunds:

"As condensates are now thought also to have played a role in the emergence of life from inanimate matter, providing reactive compartments, studying chemical reactions in the presence of condensates is a significant topic of interest."

We have changed the sentence to clarify the subclause. It now reads:

"As condensates are now thought also to have played a role in facilitating chemical reactions during the emergence of life from inanimate matter, studying chemical reactions in the presence of condensates is a significant topic of interest."

- Could you make it clear that coacervates and condensates are used interchangeably? It would be interesting to include a discussion on the differences in simulating complex coacervates and simple coacervates like the ones in this paper.

We apologise for this oversight. We have now ensured that the systems are correctly described consistently as condensates throughout. We consider that a wider discussion of the challenges of simulating simple/complex coacervates to be outside the scope of this work, but like to point out that complex coacervates can also be simulated with Martini, as demonstrated in a number of other studies (see, for example, Tsanai, M. et al. Coacervate Formation Studied by Explicit Solvent Coarse-Grain Molecular Dynamics with the Martini Model. *Chem. Sci.* **2021**, *12* (24), 8521–8530; Liu, Y. et al. Capturing Coacervate Formation and Protein Partition by Molecular Dynamics Simulation. *Chem. Sci.* **2023**, *14* (5), 1168–1175). We see no reason why Reactive Martini could not be applied equally well to simulate reactions occurring in such systems.

Results and Discussion

- Some sentences are written in a very casual style, assuming the reader is very familiar with the topic and the project. For example

"Recent work by the Spruijt group has shown"

"Parameters for unreactive benzene-1,3-dithiol were taken from the original reactive Martini paper"

We hope the clarification in the text of these examples are now less casual:

"Recent work has shown that it is possible"

"The simulation input topology for unreactive benzene-1,3-dithiol was taken from the original reactive Martini paper"

- As you say, "the number of decay modes required to fit the data are descriptive of the number of diffusion regimes present". Please provide a measure of the quality of the fitting for each case (R2 if suitable).

The fitting statistics for both single and double exponential fits are summarised in table S2, and has been detailed with with additional sentence in the main text:

The data were fitted with both single and weighted double exponential decays using Imfit, and the best model for the fit was determined by comparison of the Akaike information criterion³⁹

- The discrepancy between experimentally and computationally obtained water content is quite large (60% Vs 17%) and I do not understand how it does not compromise the analysis of the effect of the coacervate environment. I would assume the dielectric constant of the medium to be relevant for the disulfide formation, and then be very different if you consider 60 or 17% of water.

To address this, I suggest: including experimental data for this reaction to show the quality of the simulation's prediction. If that is not possible, include a comparison of simulation and experiment for a second system for which it is possible - the reaction is slower, or that can be monitored with fluorescence microscopy. Since you focus on the effect over the reaction rate, a comparison to experimental data is really missing. Alternatively, please include simulations with other phase-separating peptides and perform reactive Martini with the one with the best water content simulation data.

We recognise the reviewer's concern with regard to the water content of the condensates in the simulations. Unfortunately, we did not manage to get the experimental setup such as to actually monitor the reaction rates, and leave this for future investigations.

Following your suggestion and to further confirm the phenomena found in the manuscript, we performed additional simulations with the pH-dependent phase separating WGR-1 peptide from Baruch Leshem, A., Sloan-Dennison, S., Massarano, T. *et al.* Biomolecular condensates formed by designer minimalistic peptides. *Nat Commun* **14**, 421 (2023). Firstly, we simulated a condensate of these peptides to measure the water content, which we show in figure S10 that the water weight proportion inside the bulk of the condensate is substantially increased compared to the LFssFL system, at around 53%.

Further, we simulated these peptides together with reactive components at three different weight proportions, and again found that there was an effect regarding the size of rings found in increasing condensate concentrations. We have noted this in the results section:

This effect of condensate weight proportion was also observed in a second condensate system, using the WGR-1 peptide designed by Leshem *et al*⁴¹, results for which are shown in figure S10

- In the Partitioning section, I could not find the experimentally measured Kp for the 1,3-benzenedithiol. Please include all in a table - measured and calculated Kp and free energy of all compounds studied (including macrocycles).

It was not possible to experimentally measure the partitioning of benzene-1,3-dithiol on its own, which is why we offered the comparison of the monomer XGLKFK as the precursor to the macrocycle formation. We have now added a table (table S5) summarising all the partition coefficients we have measured.

- I did not understand the reasoning behind choosing the XGLKFK as a second guest molecule for the coacervates - a more similar variant of the benzenedithiol would be more helpful to understand what factors influence partitioning. And more reaction models would be necessary to establish a clear correlation between partitioning coefficients and reaction rate.

XGLKFK is included because it is the building block of the rings studied experimentally.

- the discussion about Figure 2 is not clear (starting on "In the previous setup..."), the naming of the six experiments could also be clearer. Since this is one of the two main conclusions of the manuscript, it needs to be indicated whether the difference between the product distributions is statistically significant. The results with water, 500, 1500 and 2500 peptides seem equivalent and I can't tell what variation is significant in MD simulations.

We have now clarified the naming of the different systems simulated according to the concentration of peptide. Regarding the significance of the difference in distributions, these datasets (of < 15 points each) are too small to be able to conduct any meaningful significance testing. We would also mention that we only make limited claims (e.g. "*The concentration of peptides added to the system only had a slight effect*"), and so are not claiming any substantial difference between the resulting distributions in these simulations.

- does the benzenedithiol reaction affect the phase separation of the peptide? Or does phase separation take place the same way in your co-formation simulations (500, 1500 and 2500 peptides) as in your non-reactive Martini ones?

The benzenedithiol reaction does not affect the phase separation of the peptides. Firstly, the reactive aspect of reactive Martini is self-contained, such that reactive components solely interact with other reactive molecules. Secondly to confirm this, we have run a non-reactive simulation with identical system composition, the final snapshot of which is shown below. The change in potential energy, as an indication of system equilibrium, shows that the two systems converge to equilibrium at approximately the same rate, further demonstrating that the reaction does not affect the phase separation. The figure below is now added in the SI (Figure S5), and we have noted it in the main text:

The progress of the reaction was not observed to affect the formation or structure of the condensate (Figure S5).

- the interfacial partitioning is an interesting finding, please discuss whether it affects the reaction. Could it explain the difference observed for pre-partitioned dithiol and co-formation simulations - I noticed the size of the condensate varies on each case, and so does the interfacial area.

There is no direct evidence from the simulations that the reaction rate is affected by the interfacial partitioning. We believe the reference to condensate size here is a reference to Figure S4, which shows before and after snapshots of each simulation condition. In the case of the pre-partitioned system, the size of the condensate has been carefully constructed to reflect the size and composition of the spontaneously formed one at 6% peptide weight. Any apparent difference from the figures is likely due to rendering. We have remade this figure (and all others throughout the text) to include scale bars for all systems to show this.

- Could you clarify if the monomer and products count refer to the whole box, or only to the coacervate phase? Please provide the values of the count and the reaction rate on a table, separating total, dilute phase and coacervate phase - this is important to give a full picture of the reaction progress.

The counts given refer to the compositions across the whole simulation box. We have now summarised the final ring counts shown in Figure 2c in table S4, and added reaction rates in table S5, denoting which part of the system they correspond to. Please note that in the co-formation simulations, where the condensate phase separates at the same time as the reaction progresses, it is not possible to precisely distinguish where any given reaction takes place, because a phase separated region may or may not have developed around reactive molecules.

- When you say "In principle, the observed shift toward larger ring sizes could have a number of reasons", you list three possibilities - concentration, peptide catalysis and thermodynamic stability of the products. You attribute the effect entirely to concentration, but it's difficult to agree

with that unless you include other coacervate-reaction pairs, or test for the other factors too - is that something you can see with reactive Martini?

Please see our above response regarding the water content of the simulations.

- Since the coacervate-forming peptide is a disulfide itself, do you observe hybrid products from the thiol-disulfide exchange between LFssFL and benzenedithiol experimentally? Or do you take it into account computationally? This might affect effective concentrations, or the assumption that the peptide is non reactive.

- I think a big question in the field is not only what happens to reactions inside coacervates, but what happens to kinetics when the reactants are involved in phase separation. Can you include a simulation of the coacervation and the reaction for LFssFL alone, starting from LF-SH? This will be a natural expectation of the active coacervates community. Otherwise, please discuss the challenges involved in making that step.

To take the previous two points together, we do not assume that the peptide is reactive, and so the reviewer is correct in that we assume that the peptide is non reactive. We are unaware of any experimental evidence that the disulfide bond in the peptide synthon so readily forms and breaks in the way that the reviewer describes. As we have also discussed in our comment on the reversibility of reactions, disulfide bonds have a high dissociation energy. It would seem unlikely that a mixture of fully synthesised peptides together with labile monomers would spontaneously break apart and subsequently form cross products. However, in principle Reactive Martini could be used to simulate the kinetics of phase separation involving the reactants, i.e. during out-of-equilibrium conditions, and this is one of the benefits of the method as mentioned before. We leave exploring this avenue for future work.

Indeed, upon mixing LFssFL experimentally with precursors of replicators (3mer and 4mer), no significant exchange was observed between the disulfides of replicator macrocycles and those of LFssFL. However, replicator building blocks (1mer) can form hybrid products with LFssFL. To prevent the formation of side products, we peroxidized 1mer to generate precursors (3mer and 4mer) while retaining a minimal amount of 1mer to facilitate thiol-disulfide exchange.

Conclusions

- In general, the conclusions are too focused on further developments and should elaborate more on the findings of the paper itself.

We apologise for this oversight. The first paragraph of conclusion now reads:

In this work, we have demonstrated how reactive Martini can be used to explore disulfide bridge formation between benzene-1,3-dithiol monomers leading to the formation of macrocycles inside biomolecular condensates. We firstly demonstrated that the Martini force field can be

used to reproduce the expected pH-dependent LLPS of a synthetic peptide. Secondly, we showed that reactive molecule monomers preferentially partition inside condensates, with partitioning coefficients comparing well between simulation and experiments. Finally, we observed that the condensates formed by the synthetic peptides can change both the speed and the outcome of the reactions between benzene-1,3-dithiol molecules by acting as an attracting environment for both the reactants and the products. This effect is dependent on the volume fraction of condensate-forming peptides in the system, which drives the formation of larger macrocycles at higher fractions.

- In the conclusion "We observed that the condensate can change both the speed and the outcome of the reactions by acting as an attracting environment for both the reactants and the products", can you explain why the partitioning of the products plays a role in the reaction rate? I did not find that claim in the discussion.

To keep this argument in line with the arguments in the discussion, we have removed the reference to the products in the sentence. It now reads:

Finally, we observed that the condensates formed by the synthetic peptides can change both the speed and the outcome of the reactions between benzene-1,3-dithiol molecules by acting as an attracting environment for the reactants

Methods

- If you are measuring the volume of the dilute phase precisely, why do you consider it in the equation that the total volume and the dilute phase volume are the same? Or could you include a table with the volume measurements?

Considering the volume of the condensate phase represents only around 1.5-2% of the total volume, we equate the total volume with the volume of the dilute phase in our calculations. The table below presents the volume measurements obtained:

replicates	system	Total volume (uL)	Supernatant volume (uL)
#1	1mer	440	430
#2	1mer	440	435
#3	1mer	440	434
#1	3mer/4mer	440	433
#2	3mer/4mer	440	428
#3	3mer/4mer	440	430

To enhance

precision, we accounted for the phase fraction of the condensate phase. Consequently, the equations were adjusted as follows:

$$\text{For 1mer: } V_{\text{Super}} = 98.4\% V_{\text{total}}; C_{\text{cao}} = (C_{\text{total}} - 0.984 C_{\text{super}}) * (V_{\text{total}} / V_{\text{coacervate}})$$

$$\text{For 3mer and 4mer: } V_{\text{Super}} = 97.8\% V_{\text{total}}; C_{\text{coa}} = (C_{\text{total}} - 0.978 C_{\text{super}}) * (V_{\text{total}} / V_{\text{coacervate}})$$

As a result, the partitioning of different species was modified as follows:

	partitioning
1mer	5.71 ± 0.65
3mer	26.2 ± 1.3
4mer	6.99 ± 1.2

These adjustments afford greater precision in our analysis and ensure that the calculations accurately reflect the dynamics of the system.

Reviewer #2 (Remarks to the Author):

Brasnett et al. investigate the impact of biomolecular condensate formation on the rate of chemical reactions using the reactive Martini model. Specifically, they explore how condensates composed of simple peptides, previously studied through experiments, can accelerate macrocycle formation. The manuscript is well-written, clear, and concise. The experimental data (i.e., spectral data) do support the findings in the paper.

Key Finding:

1. The concentration of reactive molecules within the condensates exerts the strongest influence on the size of formed macrocycles. These results carry significant implications for understanding the role of condensates in reaction kinetics.

We are grateful to the reviewer for their kind comments about the manuscript.

Minor Comments:

1. Additional details or annotations in Figure 1 would enhance its accessibility to a broad audience.

We have added some additional annotations to Figure 1, we hope that it achieves the aim of increasing its accessibility.

2. Reporting the mass density or concentration of components in Table S2 would be beneficial.

We have now added these numbers to the table.

3. Including mean and standard deviation values in Figure 2c would improve the clarity of trends.

Error bars have now been added to Figure 2c.

4. In the conclusion section, a comment comparing the reported rates to experimental findings would provide better contextualization of the key findings.

The first paragraph of the conclusion now reads to try and contextualise findings better:

In this work, we have demonstrated how reactive Martini can be used to explore disulfide bridge formation between benzene-1,3-dithiol monomers leading to the formation of macrocycles inside biomolecular condensates. We firstly demonstrated that the Martini force field can be used to reproduce the expected pH-dependent LLPS of a synthetic peptide. Secondly, we showed that reactive molecule monomers preferentially partition inside condensates, with partitioning coefficients comparing well between simulation and experiments. Finally, we observed that the condensates formed by the synthetic peptides can change both the speed and the outcome of the reactions between benzene-1,3-dithiol molecules by acting as an attracting environment for both the reactants and the products. This effect is dependent on the volume fraction of condensate-forming peptides in the system, which drives the formation of larger macrocycles at higher fractions.

Reviewer #3 (Remarks to the Author):

In this work, titled "Capturing chemical reactions inside biomolecular condensates with reactive Martini simulations" the authors present a proof-of-concept study into the application of the Martini 3 model to the formation of macrocycles of benzene-1,3-dithiol inside a condensate made of synthetic peptides. I really enjoyed reading this manuscript. I think the work represents a valuable contribution to the field. It is very exciting to see the first steps in developing and adapting the powerful Martini model towards probing chemical specificity and chemical reactions inside condensates. I would be happy to recommend publication of this work, but have a few comments and questions for the authors:

We thank the reviewer for their generous comments about our manuscript.

1. The authors explore, qualitatively, the impact of pH (low versus high pH) on the phase behaviour of LFssFL solutions with Martini 3. To mimic the effects of pH, they change the charge of the peptides to account for the change in the protonation state at low and high pH; then they run DC simulations to calculate the density profiles across the long side of the box in the slab geometry. These profiles show phase separation of the solution into a LFssFL-rich and water-rich coexisting phases at high pH, and a homogeneous solution at low pH, in perfect

agreement with the experiments of the Spruijt group. However, the approximation used to mimic the impact of pH is too crude, albeit likely necessary since accounting for the impact of pH on simulations of biomolecular phase separation is extremely challenging—e.g. pH changes are expected to yield a multiplicity of interconverting molecular states with varying protonation states. I think the paper would benefit from an explanation of why the approximation used by the authors is reasonable. What are the difficulties with describing the impact of pH on the phase behaviour of biomolecules? What are the pKas of the dipeptides? And Why is assuming homogeneous protonation states adequate in this case?

As the reviewer suggests, the treatment of pH is currently necessarily very crude, in order to capture the high-level phase behaviour of the peptide in question. The pKa of the LFssFL synthetic peptide was calculated to be ~7.5 using two prediction servers (MolGpka, *J. Chem. Inf. Model.* 2021, 61, 7, 3159–3165; Graph-pka, *Bioinformatics*, Volume 38, Issue 3, February 2022, Pages 792–798). While a recent method has enabled some dynamic interconversion of protonation states of biomolecules for Martini (*J. Chem. Phys.* 153, 024118 (2020)), this method has not yet been implemented for N termini, so cannot be used here. Although other constant pH simulation methods are available, the focus of the present work is the combination of chemical reactions and phase separated systems, rather than the nature of the condensate formation itself, though this would certainly be an interesting area of future study.

To clarify why our approximation of homogeneous protonation states is reasonable, we have added the following text in the methods:

Note that these homogeneous protonation states were used to simplistically model the pH in order to reproduce the overall phase separation of the system.

2. The authors discuss that the Martini condensates contain ~17% of water, while experiments have reported a much higher water weight of >60%. Then, they explain that this is due to the Martini underestimating the strength of interactions between amino acids and water. After this, they introduce their results of umbrella sampling simulations to quantify the PMF of moving a benzene-1,3-dithiol molecule (and the molecule attached to peptides) from the LFssFL-rich phase to the water-rich phase. The PMFs suggest spontaneous partitioning of benzene-1,3-dithiol into the LFssFL-rich phase with a small energy difference of around -9 kJ/mol. They also observe a minimum in energy near the condensate interphase, suggesting preferential accumulation of the small molecule near the interphase. However, I would expect such an energy difference to be heavily reduced and the shape of the PMF curve to change if the interactions of LFssFL with water in the model were to be increased to form condensates with a higher water content, as suggested by experiments. Can the authors comment on this?

As the reviewer suggests and as we have now shown in figure S2, it is indeed the case that it is possible to increase the strength of the peptide-water interactions to increase the density of water inside the condensate. The density profile shows that while the strength of the interaction is increased by ~5%, the proportional weight of water inside the condensate is only increased by around 5 percentage points. However, there is a clear difference between the subsequently

measured PMFs, shown in the same figure. While the energy difference between the dense and dilute regions has not changed substantially, the shape in the interfacial region has. We have further commented on this phenomenon by adding the following text at the end of the subsection regarding the partitioning:

This is confirmed by simulations shown in Figure S2 where simulations were conducted with the peptide-water interactions slightly increased, correspondingly increasing the proportion of water inside the bulk of the condensate. The free energy difference for a single benzene-1,3-dithiol is reduced from around -9 ± 1 kJ/mol to around -7 ± 1 kJ/mol. We additionally observe a change in shape of the PMF around the interfacial region, suggesting the interfacial regions of condensates may only play a limited role in the recruitment of reactive components.

We have also aided this clarification by removing the following text from the bottom of the first paragraph in the section:

While the PMF suggests that the molecules will easily partition deep into the condensates, they also feature a slight minimum around the interfacial region, around 6 nm from the centre of mass of the slab geometry. This suggests that the interfacial region may play a significant role in the recruitment of these molecules to within the condensate

And added the following in the methods:

To increase the interaction between peptides and water for selected simulations, a virtual site was added to the backbone beads of the peptides with an extra interaction of $\epsilon = 0.465$ nm and $\sigma = 0.1$ kJ/mol to the water beads of the system, as detailed elsewhere.

3. The next part of the paper is very exciting as it demonstrates the application of Martini to probe chemical reactions inside condensates, particularly the formation of macrocycles from monomers of benzene-1,3-dithiol. However, a big drawback that I see is disclosed by the authors: their algorithm only allows for the formation, but not breakage of the macrocycles. This implies that the forces in their system are non-conservative, and the system is out of equilibrium. Thus, the preferential cycle sizes they observe (Fig 2c) and rates of reactions are simply determined by how long the simulation has been run, or can the authors describe steady states?

We are pleased the reviewer agrees that this is an exciting application of the Reactive Martini model. The Reactive Martini algorithm, however, does not prohibit bond breaking. Whether bond breaking occurs in practice depends on the depth of the energy well used to drive bond formation. The rate of bond breaking and formation can be further tuned by adding a reactive barrier, as explained in detail in the original paper. Already in the systems presented, there is evidence of transient connectivity between monomers, particularly among the components forming larger macrocycles. These larger rings are very transient (Figure S7), such that there is breakage of the macrocycles. To further address this, we repeated the simulation at 6% w/w using a smaller potential well (20 kJ/mol as opposed to 60 kJ/mol) for the reactive components

in the system. We have now presented this data in figure S7, which shows that it is indeed possible to both simulate the formation and breakage of macrocycles, and reach a steady state of larger rings. We have added the following text in the results section to detail these systems:

Another aspect that could affect the ring size distribution is the possibility for disulfide bond breaking, which is not captured in our current model. The dissociation energy of disulfide bonds is on the order of 250 kJ/mol⁴⁴, and therefore irreversible on the time scale of our simulations, but depending on pH and presence of a reductive environment disulfide bonds can be formed and broken reversibly in practice, e.g. via thiol-disulfide exchange. As a further proof-of-concept of how the reactive potential may be tuned to optimally reflect experimental conditions, we performed additional simulations to demonstrate that with a smaller potential well, bond breakage in this manner can be reproduced (Figures S7-S8). In this case too, very large macrocycles are formed, but are also seen to break apart. Similar to the irreversible case, 2mers and 3mers still remain the dominant ring sizes observed in these systems, appearing to approach an equilibrium population in the last 20 ns of simulation time.

This has also been noted in the methods section:

For simulations using a smaller potential well, the notebook provided with reactive Martini was used to write new tabulated potential files with a reactive potential reduced from 60 kJ/mol to 20 kJ/mol.

Minor:

1. The term “coacervate” is used incorrectly. Coacervate refers to a condensate formed by at least two components via associate heterotypic interactions (e.g polyR and RNA).

We apologise for this oversight, we have now referred to ‘condensates’ throughout the manuscript. Note, however, that a simple coacervate only requires a single component to undergo LLPS (as opposed to a complex coacervate).

REVIEWERS' COMMENTS:

Reviewer #1 (Remarks to the Author):

The authors have significantly improved the manuscript, with many additional results added in supplementary information. I have some minor comments left but overall can support its publication.

- In regards to the comment on the rebuttal that the manuscript "is not meant to be a study on the effect of condensates on reactions in general", could you please update the abstract where it says "To investigate how biomolecular condensates may affect the rates and nature of chemical reactions, here we perform reactive molecular dynamics simulations using the coarse-grained Martini forcefield" - causing the misunderstanding?
- The explanation about Martini is now clear.
- Fitting statistics are now included.
- Authors included a second system for the simulation of condensation – I understand the difficulties in obtaining experimental data on different systems are an external factor to the manuscript.
- You could consider, if suitable, relating your finding about the effect of condensate volume fraction to the preprint by Spruijt et al (<https://chemrxiv.org/engage/chemrxiv/article-details/66258e5c21291e5d1d54e51d>), where they find an optimal volume fraction to increase the rate of bimolecular reactions?
- Reversibility of the reaction is now taken into account.
- Writing style has been improved where pointed.
- The role of the condensate-forming peptide's reactivity is left for another paper, which I understand. You could however include in the text (under Methods >> Experiments) what you say in the rebuttal about the monomers being peroxidized to prevent reactions with LFssFL.
- Phase volumes are now clearly presented in a table.

Reviewer #2 (Remarks to the Author):

The authors have satisfactorily addressed the points raised in the initial review. They may wish to comment on the following in their manuscript:

1. Why WGR-1 was specifically chosen for additional simulations?
2. It is interesting why XGLKFK shows a deeper minimum at the interface compared to benzene-1,3-diol (fig. S2 and S3), what may be a reason?

Reviewer #3 (Remarks to the Author):

The authors have addressed all my comments satisfactorily.

We thank all the reviewers for their additional comments on the manuscript. We have addressed their final comments below:

REVIEWERS' COMMENTS:

Reviewer #1 (Remarks to the Author):

The authors have significantly improved the manuscript, with many additional results added in supplementary information. I have some minor comments left but overall can support its publication.

- In regards to the comment on the rebuttal that the manuscript “is not meant to be a study on the effect of condensates on reactions in general”, could you please update the abstract where it says “To investigate how biomolecular condensates may affect the rates and nature of chemical reactions, here we perform reactive molecular dynamics simulations using the coarse-grained Martini forcefield” - causing the misunderstanding?

This sentence in the abstract now reads:

To demonstrate how molecular dynamics may be used to capture chemical reactions in condensates, here we perform reactive molecular dynamics simulations using the coarse-grained Martini forcefield

- The explanation about Martini is now clear.
- Fitting statistics are now included.
- Authors included a second system for the simulation of condensation – I understand the difficulties in obtaining experimental data on different systems are an external factor to the manuscript.
- You could consider, if suitable, relating your finding about the effect of condensate volume fraction to the preprint by Spruijt et al (<https://chemrxiv.org/engage/chemrxiv/article-details/66258e5c21291e5d1d54e51d>), where they find an optimal volume fraction to increase the rate of bimolecular reactions?

We thank the reviewer for this suggestion - we had also noticed this paper posted a few days after resubmission, so agree that it is a timely inclusion. We have noted this in our discussion:

Similar volume fraction rate enhancements have been found recently using kinetic network models.

- Reversibility of the reaction is now taken into account.
- Writing style has been improved where pointed.
- The role of the condensate-forming peptide's reactivity is left for another paper, which I understand. You could however include in the text (under Methods >> Experiments) what you say in the rebuttal about the monomers being peroxidized to prevent reactions with LFssFL.

We have now included the sentence from our initial response in the methods section

- Phase volumes are now clearly presented in a table.

Reviewer #2 (Remarks to the Author):

The authors have satisfactorily addressed the points raised in the initial review. They may wish to comment on the following in their manuscript:

1. Why WGR-1 was specifically chosen for additional simulations?

The WGR-1 peptide was chosen for additional simulations as another short peptide that undergoes phase separation. To help clarify this the sentence now reads:

... observed in a second small peptide-based condensate system.

2. It is interesting why XGLKFK shows a deeper minimum at the interface compared to benzene-1,3-diol (fig. S2 and S3), what may be a reason?

To clarify this, we have added the following sentence:

The minimum is likely as a result of the larger XGLKFK molecule covering a larger interfacial area, and therefore further reducing the surface tension.

Reviewer #3 (Remarks to the Author):

The authors have addressed all my comments satisfactorily.

We thank the reviewer for their work in shaping our manuscript